# NMR-Based Metabolic Profiling of the Effects of α-Ketoglutarate Supplementation on Energy-Deficient C2C12 Myotubes

**DOI:** 10.3390/molecules28093840

**Published:** 2023-04-30

**Authors:** Yantong Li, Shuya Zhang, Caihua Huang, Donghai Lin

**Affiliations:** 1Key Laboratory of Chemical Biology of Fujian Province, Department of Chemical Biology, College of Chemistry and Chemical Engineering, Xiamen University, Xiamen 361005, China; 2Research and Communication Center of Exercise and Health, Xiamen University of Technology, Xiamen 361021, China

**Keywords:** AKG supplementation, myotubes, biomolecular NMR, metabolomics, energy deficiency

## Abstract

Skeletal muscle is closely linked to energy metabolism, but it is inevitably deprived of energy. Cellular differentiation is an essential and energy-demanding process in skeletal muscle development. Much attention has been paid to identifying beneficial factors that promote skeletal muscle satellite cell differentiation and further understanding the underlying regulatory mechanisms. As a critical metabolic substrate or regulator, α-ketoglutarate (AKG) has been recognized as a potential nutritional supplement or therapeutic target for skeletal muscle. We have previously found beneficial effects of AKG supplementation on the proliferation of C2C12 myoblasts cultured under both normal and energy-deficient conditions and have further elucidated the underlying metabolic mechanisms. However, it remains unclear what role AKG plays in myotube formation in different energy states. In the present study, we investigated the effects of AKG supplementation on the differentiation of C2C12 myoblasts cultured in normal medium (Nor myotubes) and low glucose medium (Low myotubes) and performed NMR-based metabonomic profiling to address AKG-induced metabolic changes in both Nor and Low myotubes. Significantly, AKG supplementation promoted myotube formation and induced metabolic remodeling in myotubes under normal medium and low glucose medium, including improved energy metabolism and enhanced antioxidant capacity. Specifically, AKG mainly altered amino acid metabolism and antioxidant metabolism and upregulated glycine levels and antioxidase expression. Our results are typical for the mechanistic understanding of the effects of AKG supplementation on myotube formation in the two energy states. This study may be beneficial for further exploring the applications of AKG supplementation in sports, exercise, and therapy.

## 1. Introduction

Skeletal muscle generates strength and power to sustain physical activity, which is closely related to its energy state. Skeletal muscle has been previously shown to contribute to basal energy metabolism and to store many important metabolic substrates, such as amino acids and carbohydrates [1]. In skeletal muscle development, myoblast differentiation is a highly energy-demanding process that determines the final muscle formation [2]. Cellular differentiation also plays an extremely crucial role in the repair and regeneration of injured muscle [3,4]. However, the energy supply is not always sufficient due to the complexity of physiological conditions, which can lead to aberrant consequences [5]. Since regenerative capacity and muscle fiber plasticity are the typical characteristics of skeletal muscle [6], it is possible and necessary to restore the strength and function of skeletal muscle from energy deficiency. In particular, nutritional supplementation has been developed as an effective means of supplying energy to muscles and maintaining muscle mass [7,8]. In recent years, there has been a growing interest in identifying regulatory factors in the process of skeletal muscle differentiation and exploring their regulatory mechanisms [9,10].

As an important metabolic substrate or metabolic regulator, α-ketoglutarate (AKG) has been validated by extensive research as a potential nutritional supplement or therapeutic target [11]. He et al. found that by interfering with the intestinal flora, AKG can promote the differentiation of intestinal cells and slow down the development of colorectal cancer by inhibiting the WNT pathway [12]. In addition, AKG is considered a potential new target for the prevention and treatment of liver fibrosis [13]. Additionally, Tatara et al. found that adding AKG to the diet of piglets before and after weaning can effectively promote the formation of bone tissue [14]. Importantly, AKG has shown great potential to reduce muscle wasting [15,16] and improve muscle condition [17]. Our previous work has shown that AKG can promote C2C12 myoblast proliferation in low glucose medium [18] and reduce C2C12 myotube atrophy in no-glucose medium [19]. However, the influence of AKG on myotube formation in different energy states and the underlying mechanisms remain unclear. Therefore, it is essential to further investigate the long-term effects of AKG on C2C12 myotubes, which are more representative of skeletal muscle under physiological conditions than myoblasts.

As metabolites serve as direct signatures of biochemical activity [20], metabolomics analysis has been used to systematically elucidate the molecular mechanisms of nutritional supplements [13,21]. Meanwhile, ^1^H nuclear magnetic resonance (NMR) has been identified as a robust technique for metabonomic applications due to its prominent advantages [22,23]. Here, we employed NMR-based metabolomics to investigate the metabolic effect of AKG on myotube formation and attempt to elucidate the potential mechanism. We used normal and low glucose conditions to simulate two different physiological states and elaborated the metabolic profile of aqueous metabolites in C2C12 myotubes after AKG treatment by cellular metabolomic analysis. Our results show that AKG facilitates myotube formation and induces cellular metabolic remodeling, suggesting that the beneficial effect may be closely related to energy metabolism and antioxidant effects. This research suggests that AKG supplementation would be an attractive strategy for rescuing skeletal muscle from energy deficiency and provides some important insights for improving muscle conditions in sports and therapy.

## 2. Results

### 2.1. AKG Supplementation Promoted Myoblast Differentiation

C2C12 cells were divided into four groups according to the different culture conditions. Cells cultured in the normal medium with and without AKG supplementation were grouped as Nor-A and Nor, whereas those cultured in the low glucose medium with and without AKG supplementation were grouped as Low-A and Low. As expected, the number of myotubes in the low glucose medium was less than in the normal medium over the same period. In addition, the cell fusion rate in low glucose medium was significantly increased by AKG, and Low-A myotubes were longer than Low myotubes from day 2 to day 6 (Figure 1A and Appendix A). Compared to Nor cells, more Nor-A cells tended to fuse to form myotubes at earlier stages. However, the number and length of Nor and Nor-A myotubes began to converge on day 6. Myotubes cultured in low glucose medium had significantly decreased expression of myosin heavy chain (MYH) compared to cells cultured in normal medium. Further analysis showed that the decreased MYH expression in low glucose conditions was restored by AKG supplementation, whereas there was no significant difference between the Nor and Nor-A groups (Figure 1B,C).

### 2.2. NMR Spectra of Aqueous Extracts of C2C12 Myotubes

Typical 1D ^1^H-NMR spectra recorded on aqueous extracts from the four groups of C2C12 myotubes are shown in Figure 2. A total of 32 metabolites were assigned and summarized in Appendix A. The resonance assignments of the metabolites were validated by 2D ^1^H–^13^C HSQC and ^1^H–^1^H TOCSY spectra (Appendix A). By visual inspection of the NMR spectra, we found significant accumulations of intracellular AKG in the Nor-A group (Appendix A) but not in the Low-A group (Appendix A), suggesting that AKG was utilized more by the latter.

### 2.3. Multivariate Statistical Analysis of the Metabolic Profiles of Myotubes

We further performed multivariate statistical analysis on the NMR spectral data to explore the metabolic profiles of the three groups of myotubes. First, we established unsupervised principal component analysis (PCA) to overview trends in sample grouping and metabolic separation for the four groups of myotubes (Figure 3A–C). Myotubes cultured in low glucose medium were metabolically distinct from those cultured in normal medium. AKG supplementation profoundly altered the metabolic profiles of myotubes, regardless of the culture condition.

We then performed supervised orthogonal partial least-squares discriminant analysis (OPLS-DA) to optimize the metabolic distinctions between the groups. The OPLS-DA score plots (Figure 3D–F) show clear metabolic distinctions between the groups. We performed a response permutation test (*n* = 200) to evaluate the reliability of the OPLS-DA model (Figure 3G–I). The obtained cross-validation plots show the following Q2 and R2 values: R2 = 0.999, Q2 = 0.998 for Low vs. Nor; R2 = 0.972, Q2 = 0.754 for Low-A vs. Low; and R2 = 0.846, Q2 = 0.768 for Nor-A vs. Nor, indicating that these established OPLS-DA models were reliable.

### 2.4. Identification of Differential Metabolites and Significant Metabolites

To identify differential metabolites whose levels differed significantly between groups, we calculated the relative concentrations of the assigned metabolites based on the integrals of the metabolites relative to the sum integrals (Appendix A). We then performed an independent sample *t*-test (two-tailed) with a criterion of statistical significance of *p* < 0.05 to identify differential metabolites (Figure 4). A total of 27 differential metabolites were identified for the pairwise comparison of Low vs. Nor, including 19 increased metabolites (leucine, isoleucine, valine, acetate, glutamate, pyroglutamate, aspartate, lysine, creatine, phosphocreatine, beta-alanine, GPC, taurine, tyrosine, phenylalanine, histidine, NAD+, formate, and AXP), and eight decreased metabolites (ethanol, alanine, glucose, glycine, lactate, threonine, UDP-glucose, and NADH). Moreover, five differential metabolites were identified for the pairwise comparison of Low-A vs. Low, including three increased metabolites (pyroglutamate, glycine, and 1-methylhistidine) and two decreased metabolites (taurine and lactate). Furthermore, 16 differential metabolites were identified for the pairwise comparison of Nor-A vs. Nor, including 11 increased metabolites (AKG, aspartate, creatine, phosphocreatine, GPC, taurine, glycine, GTP, 1-methylhistidine, formate, and AXP) and five decreased metabolites (alanine, glutathione, glutamate, myoinositol, and threonine).

We identified significant metabolites with a criterion of variable importance in projection (VIP) > 1, which were calculated from the OPLS-DA models (Figure 5). Strikingly, glycine had the highest VIP score in every pairwise comparison. With two criteria of *p* < 0.05 and VIP > 1, we identified 8, 3, and 6 characteristic metabolites for pairwise comparisons of Low vs. Nor, Low-A vs. Low, and Nor-A vs. Nor, respectively (Table 1).

### 2.5. Significantly Altered Metabolic Pathways of Myotubes with AKG Supplementation

We then performed metabolic pathway analysis to identify significantly altered metabolic pathways (termed significant pathways) of myotubes based on the relative concentrations of assigned metabolites (Table 2 and Appendix A). A total of 10 significant pathways were identified in the Low group compared to the Nor group (Appendix A), as follows: P1, alanine, aspartate, and glutamate metabolism; P2, glycine, serine, and threonine metabolism; P3, glutathione metabolism; P4, D-glutamine and D-glutamate metabolism; P5, beta-alanine metabolism; P6, taurine and hypotaurine metabolism; P7, phenylalanine metabolism; P8, phenylalanine, tyrosine, and tryptophan biosynthesis; P9, nicotinate and nicotinamide metabolism; and P10, histidine metabolism. Compared with normal controls, the impaired metabolic pathways mainly involved amino acid metabolism, energy metabolism, and antioxidant metabolism. In addition, AKG supplementation significantly altered four significant pathways in energy-deficient myotubes, including P2, P3, P6, and P10 (Appendix A), but six significant pathways in normal myotubes, including P1–4, P6, and P10 (Appendix A).

### 2.6. Expressions of Proteins Corrected with Energy Metabolism and Antioxidant Capacities of Myotubes

To investigate how AKG affects myotube formation, we first detected proteins correlated with energy metabolism and ATP levels. Since AKG is the substrate that can supply energy to cells, we measured the expression of AMP-activated protein kinase (AMPK), which is the key sensor in the regulation of energy metabolism [24]. Our results showed that the AMPK phosphorylation level (p-AMPK/AMPK) of the Low group was significantly increased relative to the Nor group (Figure 6A,B). It is worth noting that AKG significantly decreased the ratio of p-AMPK to AMPK in myotubes under low glucose conditions, indicating the critical effects of AKG on energy metabolism in the energy-deficient state. However, compared to the Nor group, the AMPK phosphorylation level of the Nor-A group did not change significantly, implying that myotubes had reached a stable state by day 6. Interestingly, treatment with AKG appeared to increase ATP levels regardless of culture condition (Figure 6G).

We next investigated whether the beneficial effect of AKG on myotubes was related to its antioxidant function. As shown in Figure 6E, the Low group exhibited significantly elevated levels of reactive oxygen species (ROS) compared with the Nor group, indicating the abnormal accumulation of ROS in myotubes. Importantly, AKG supplementation profoundly eliminated the accumulated intracellular ROS. We also measured total antioxidant capacities (T-AOCs) in the four groups of myotubes (Figure 6F) and found that AKG fully restored the decreased T-AOC in the low glucose state. Moreover, we also determined the expressions of catalase (CAT) and superoxide dismutase (SOD), which play key roles in cellular antioxidation. The CAT levels showed no significant changes between the Nor and Nor-A groups, whereas they showed an evident increase with the addition of AKG in the low glucose medium (Figure 6A,C). Compared with the Nor group, low glucose cultures significantly increased SOD levels, suggesting that long-term oxidative stress could increase SOD levels. Notably, AKG increased SOD levels to a higher level under low glucose conditions (Figure 6A,D). Taken together, these results suggest that the effect of AKG supplementation on promoting myotube formation is closely related to energy metabolism and antioxidant capacity.

To illustrate the role of AKG supplementation in myotube formation, we have drawn a schematic representation (Figure 7) based on KEGG and the MetaboAnalyst webserver. In general, AKG seems to be a rescuer for C2C12 myotubes, reversing a number of impairments caused by energy deprivation.

## 3. Discussion

Skeletal muscle has been identified as the largest organ in the body [25] and a key regulator of whole-body energy metabolism [26]. It is now well established that the energy status of skeletal muscle affects its size and function. Unfortunately, it is inevitable that muscles will experience energy deficiency due to various physiological conditions. Several studies have investigated the factors that affect skeletal muscle. Recently, researchers have investigated the effects of AKG on muscle, and as mentioned above, we conducted a preliminary investigation on C2C12 myoblasts in previous work. Cellular differentiation is not only indispensable but also an energy-demanding process in skeletal muscle development. However, the metabolic changes induced by AKG in myotube formation and the underlying molecular mechanisms are not fully understood. In the present work, we used two culture media, normal and low glucose, to simulate the different conditions that may occur during skeletal muscle differentiation and continued to supplement 2 mM AKG during myotube formation. Notably, our results have demonstrated the beneficial effects of AKG supplementation on myotube formation and elucidated the potential mechanisms involved.

### 3.1. AKG Induces Metabolic Remodeling and Promotes Myotube Differentiation

Both myotube formation and myoblast fusion had been performed under serum starvation conditions [27], which in this experiment was a situation of energy deficiency. Our results showed that the metabolism of the myotubes showed significant discrepancies under the two energy conditions, indicating that the glucose level has a huge impact on the metabolism of skeletal muscle, as glucose is the main energy source. In agreement with previous results [28], we found that myotube differentiation was slower in the low glucose culture. Remarkably, AKG supplementation accelerated myotube formation under both energy conditions. This finding broadly supports the work of other studies in this area, which have documented that AKG can promote skeletal muscle hypertrophy by promoting protein synthesis [29] and reducing corticosterone-induced protein degradation in skeletal muscle cells [16]. Interestingly, only a small difference was observed between Nor-A cells and Nor cells when myotube differentiation was complete under normal culture. However, the cells continued to differentiate under low glucose culture on day 6, demonstrating that AKG has a significant effect on promoting myotube formation in the energy-deficient state. This may indicate that fully functional myotubes can self-regulate to cope with long-term intervention of AKG in normal cultures. Under low glucose conditions, where the cells are severely starved of energy, AKG supplementation is effective in supporting cell differentiation.

### 3.2. AKG Promotes Energy Metabolism and the Antioxidant Capacity of Myotubes

As an intermediate in the tricarboxylic acid (TCA) cycle, the energetic importance of AKG is obvious. In this work, we used normal and low glucose cultures to simulate two different energy conditions. In the normal culture, AKG supplementation obviously increased the levels of intracellular creatine and creatine phosphate, suggesting that AKG supplementation can provide more energy supply by promoting the levels of creatine and creatine phosphate in cells during the differentiation process [30]. However, the addition of AKG to the low glucose differentiation process did not change the levels of creatine or creatine phosphate. A possible explanation is that under conditions of energy starvation, the regulation of creatine and creatine phosphate by the myotube has been activated, and therefore AKG cannot adjust the levels of creatine and creatine phosphate to supply more energy. In this situation, AKG may be more inclined to be a substrate to produce more ATP through the TCA cycle and oxidative phosphorylation. In addition, the significantly lower level of phosphorylated AMPK in the Low-A group compared to the Low group also confirmed our hypothesis. An interesting finding is that ATP levels were higher in energy-deprived myotubes, which seems puzzling but has also been reported in previous studies [31,32]. Extensive research has shown that under ATP depletion, AMPK kinase is phosphorylated at Thr172 by upstream kinases [33,34], which is inconsistent with our results in the Low and Low-A groups (Figure 6A,B,G). Furthermore, the above process leads to the activation of AMPK, which in turn suppresses ATP-consuming anabolic processes and stimulates catabolism and ATP generation [33].

The fact that myotubes are exposed to oxidative stress under extreme energy starvation has been reported [35]. At that time, taurine metabolism was an important way for myotubes to resist oxidative stress [36,37]. Our previous work has demonstrated the protective effects of taurine on cisplatin-impaired myoblasts [38]. In myotubes cultured under normal conditions, AKG supplementation reduced glutathione levels but increased glycine and taurine levels, suggesting that AKG promotes taurine synthesis during long-term differentiation. Notably, since glutathione and taurine share the same precursor, taurine synthesis may influence GSH synthesis [37]. Compared to normal myotubes, glycine was profoundly decreased and taurine was significantly increased in myotubes differentiated under long-term low glucose conditions. These results showed that the myotubes initiated the antioxidation to respond to extreme starvation, which is in agreement with our previous research [18]. Interestingly, we found that intracellular glycine levels were upregulated by AKG supplementation under both normal and low glucose conditions. Glycine is one of the synthetic sources of glutathione or taurine [39], suggesting an effect of AKG on the regulation of cellular antioxidant metabolism. Another important finding was that taurine levels in Low-A myotubes were lower than in Low myotubes but higher than in Nor myotubes. A possible explanation for this could be that AKG partially alleviates the stress response of myotubes caused by harsh conditions. Accordingly, our study showed that low glucose treatment dramatically increased intracellular ROS in myotubes, which was fully reversed by AKG treatment. Encouragingly, the measurements of CAT and SOD expressions and total antioxidant capacity also supported the above ideas. These results are consistent with the findings of several previous works on the antioxidant function of AKG [11,40,41]. Despite these promising findings, questions remain. In the present work, we have just preliminarily explored the potential mechanisms of the beneficial effects of AKG on myotube formation. Further research should be undertaken to investigate and validate the specific mechanism, such as detecting changes in metabolic enzymes in metabolic pathways.

## 4. Materials and Methods

### 4.1. Cell Culture

The murine skeletal muscle cell line C2C12 was obtained from the China Center for Typical Culture Collection (CCTCC; Wuhan, China). Dulbecco’s modified Eagle’s medium with glucose (Nor; HyClone, Logan, UT, USA) or without glucose (No; Gibco, Carlsbad, CA, USA) supplemented with 10% (*v*/*v*) fetal bovine serum (Gibco, Carlsbad, CA, USA), 100 U/mL penicillin, and 100 mg/mL streptomycin were used as the growth medium (GM) for cell passage. Differentiation medium (DM) was supplemented with 2% (*v*/*v*) horse serum (Gibco, Carlsbad, CA, USA), 100 U/mL penicillin, and 100 mg/mL streptomycin. Low glucose medium was mixed with DMEM Nor and DMEM No in a ratio of 1:8. Cells were cultured in a humidified incubator containing 5% (*v*/*v*) CO_2_ at 37 °C. α-Ketoglutarate suitable for cell culture was purchased from Sigma, and the supplemental concentration was 2 mM, which was determined by a primary experiment. C2C12 cells were cultured in the differentiation medium for myotube formation, and the medium was changed every two days. AKG was supplemented from day 1 to day 6 for 6 days.

### 4.2. Cell Morphological Images

C2C12 cells were washed three times with PBS to remove the dead cells. Cell morphological images were then taken randomly using a fluorescence microscope (Motic, Xiamen, China).

### 4.3. Western Blotting Analysis

RIPA buffer (Sangon Biotech, Shanghai, China) containing protease and phosphatase inhibitors was used to obtain cell lysates. The cell lysates were then loaded onto a sodium dodecyl sulfate-polyacrylamide gel after the addition of loading buffer. The gel was then transferred to PVDF membranes (GE, Freiburg, Germany). Membranes were blocked with 5% non-fat milk at room temperature for 60 min and incubated with primary antibodies overnight at 4 °C while being shaken. The membranes were then incubated with secondary antibodies for 1 h at room temperature. The signal was detected using commercially available enhanced chemiluminescence reagent (ECL, Beyotime, Shanghai, China) and chemiluminescence equipment (Clinx, Shanghai, China). Antibodies used were GAPDH (Proteintech, Wuhan, China), β-actin (Proteintech, Wuhan, China), MYH (Santa Cruz Biotechnology, Dallas, TX, USA), CAT (Proteintech, Wuhan, China), SOD (Proteintech, Wuhan, China), p-AMPK (CST, Boston, MA, USA), and AMPK (CST, Boston, MA, USA).

### 4.4. Intracellular Metabolite Extraction and NMR Sample Preparation

Aqueous metabolites were extracted from C2C12 myotubes for NMR analyses according to the previously described protocol [22]. Briefly, cells were rapidly rinsed three times with cold phosphate-buffered saline (PBS, pH 7.4) to reduce residual medium. Myotube metabolism was then terminated by methanol, and cells were harvested into a 15 mL centrifuge tube. Chloroform and water were then added to extract intracellular metabolites. The aqueous phase was lyophilized and used for NMR-based metabolomic analysis. The aqueous cell extract powder was dissolved in 550 μL of phosphate buffer [50 mM, pH 7.4, 100% D_2_O, 0.01 mM sodium 3-(trimethylsilyl) propionate-2,2,3,3-d4 (TSP)] and followed by centrifugation at 12,000× *g* for 15 min at 4 °C. The supernatants were transferred into 5 mm NMR tubes for NMR detection.

### 4.5. NMR Measurements

All NMR measurements were performed at 298 K on a Bruker Avance III 600 MHz NMR spectrometer equipped with a BBFO cryoprobe (Bruker Bio Spin, Rheinstetten, Germany). One-dimensional (1D) ^1^H spectra were acquired using the pulse sequence NOESYGPPR1D [RD-G_1_- 90°-t-90°-τ_m_-G_2_-90°-ACQ] with the following acquisition parameters: relaxation delay (RD), 4 s; short delay (t), 4 μs; mixing time (τ_m_), 10 ms; spectral width, 20 ppm; acquisition time, 1.93 s; and 128 transients. The methyl groups of the TSP molecule were set at 0 ppm for chemical shift calibration. The two-dimensional (2D) ^1^H–^13^C heteronuclear single quantum coherence (HSQC) spectrum was recorded with a spectral width of 10 ppm in the ^1^H dimension and 110 ppm in the ^13^C dimension, a data matrix of 1024 × 256 points, and a relaxation delay of 1.5 s. The 2D total correlation spectroscopy (TOCSY) spectrum was recorded with a spectral width of 10 ppm in both ^1^H dimensions, a data matrix of 2048 × 256 points, and a relaxation delay of 1.5 s.

### 4.6. NMR Data Preprocessing

All 1D ^1^H-NMR spectra were adjusted by phase correction, baseline correction, and resonance alignment using MestReNova 9.0 software (Mestrelab Research S.L., Santiago de Compostela, Spain). The 1D ^1^H spectral regions of 9.5–0.6 ppm were segmented into bins with a width of 0.01 ppm for further multivariate statistical analysis. The water region of 5.2–4.7 ppm was excluded to eliminate the distortion from the residual water resonance in all 1D ^1^H spectra. The peak integrals of the segments were normalized by the sum of the peak integrals to compensate for possible variations in sample concentrations. The sum of the peak integrals was set to 100 for each spectrum. The normalized integrals were used to represent the relative concentrations of the assigned metabolites. Singlet or non-overlapping resonances in each NMR spectrum were selected to calculate metabolite integrals for pairwise comparison between groups.

### 4.7. Metabolomic Analysis

For multivariate statistical analysis, the 1D ^1^H-NMR spectral data of myotube extracts were performed using SIMCA-P software (version 12.0.1, Umetrics, Umea, Sweden). Pareto scaling was applied to the normalized NMR spectral data to enhance the significance of low-abundance metabolites without increasing noise. PCA was used to illustrate metabolically grouping trends of myoblasts and show clustering for the NMR dataset. OPLS-DA was further used to confirm the grouping trends and improve metabolic separation between groups.

The robustness and stability of the OPLS-DA model were measured by cross-validation with a response permutation test (200 times). Parameters R2 and Q2 were calculated to indicate the interpretability and predictability of the model, respectively. The reliability of the model increased when the R2 and Q2 values approached 1. Metabolites were assigned using the Chenomx NMR Suite (version 8.1, Chenomx Inc., Edmonton, Canada) and the Human Metabolome Database (HMDB, http://www.hmdb.ca/ (accessed on 4 February 2021)). In addition, 2D ^1^H–^13^C HSQC and ^1^H–^1^H TOCSY spectra were recorded to aid in metabolite assignment.

Relative concentrations of the assigned metabolites were calculated based on their integrals normalized to the sum of all integrals. Univariate data analysis was used to quantitatively compare the relative concentrations of metabolites between groups using a *t*-test with GraphPad Prism software (version 8.0, San Diego, CA, USA). Data are expressed as mean ± SD. Metabolites with a statistical significance of *p* < 0.05 were identified as differential metabolites.

Based on the relative concentrations of the assigned metabolites, metabolic pathway analysis was performed on the MetaboAnalyst 5.0 webserver (https://www.metaboanalyst.ca (accessed on 6 February 2021)). Significantly altered metabolic pathways were identified by a combination of metabolite set enrichment analysis (*p* < 0.05) and pathway topological analysis (pathway impact value > 0.2).

### 4.8. Measurement of Intracellular ATP Level

The medium was removed, and the cells were washed twice with ice-cold PBS buffer. After the addition of 1.2 mL of lysis buffer, the lysates were scraped and collected in an Eppendorf tube. The Eppendorf tubes were centrifuged for 5 min at 12,000× *g*. ATP concentrations were measured according to the manufacturer’s instructions. Both the ATP content kit (Beyotime, Shanghai, China) and the BCA protein assay (Beyotime, Shanghai, China) were used to calculate the ATP concentration per mg of protein.

### 4.9. Assay of Cellular Total Antioxidant Capacity

Cells were centrifuged at 3000× *g*, and then the supernatant was discarded. A quantity of 2.4 mL of lysis buffer was added, and the lysates were centrifuged for 10 min at 12,000× *g*. Cellular T-AOC was detected using the total antioxidant capacity kit (Nanjing Jiancheng Bioengineering Institute, Nanjing, China). The protein concentration in each sample was analyzed by BCA protein assay (Beyotime, Shanghai, China) to calculate the concentration of T-AOC per mg of protein.

### 4.10. Assay of Intracellular ROS Level

Cells were digested with trypsin-EDTA (Biological Industries, Shanghai, China) and transferred to Eppendorf tubes. The Eppendorf tubes were then centrifuged for 1 min at 12,000× *g*, and the supernatant was discarded. ROS levels were then detected using the H_2_DCFDA (2′,7′-dichlorofluorescin diacetate) probe (Sigma-Aldrich, Milpitas, CA, USA). The myotubes were incubated with H_2_DCFDA (10 μM) for 30 min at 37 °C, and the fluorescence intensity of H_2_DCFDA was detected using the Multilabel Reader (Enspire, Turku, Finland).

## 5. Conclusions

Our results revealed the metabolic remodeling of C2C12 myotubes induced by AKG supplementation and confirmed the critical role of AKG during the differentiation process of skeletal muscle cells. In addition, we provided the potential molecular mechanisms underlying the beneficial effects of AKG supplementation in promoting myotube formation. Specifically, AKG promoted differentiation, stimulated amino acid metabolism and antioxidant metabolism, facilitated ROS clearance, and upregulated the expression of antioxidase, which robustly protected myotubes from energy deprivation. Prospectively, the knowledge gained from this study may be of benefit to the application of AKG as a therapeutic target or nutritional supplement.

## Figures and Tables

**Figure 1 molecules-28-03840-f001:**
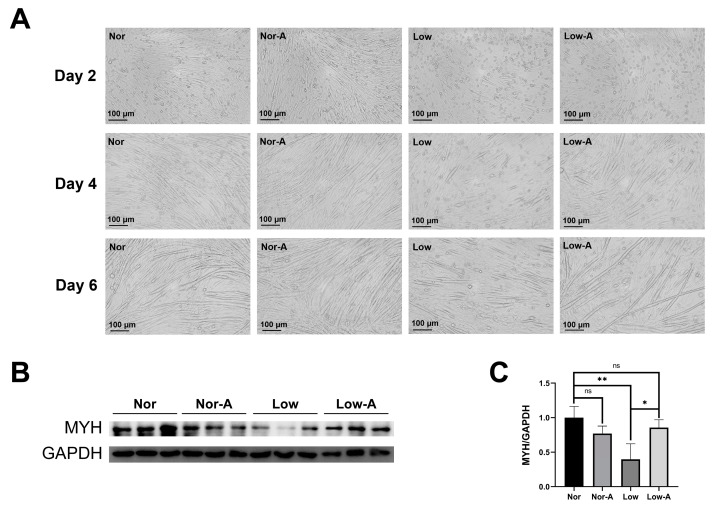
Differentiation abilities of the Nor, Nor-A, Low, and Low-A groups of C2C12 myotubes. (**A**) Morphology of the four groups of myotubes on Days 2, 4, and 6. (**B**) Western blot analysis of MYH expressions in the four groups of myotubes. The anti-GAPDH antibody was used to standardize the amount of the measured protein in each lane. (**C**) Statistical analysis of the protein expressions shown in panel B. Statistical significance: ns, *p* > 0.05; *, *p* < 0.05; **, *p* < 0.01 (*n* = 3).

**Figure 2 molecules-28-03840-f002:**
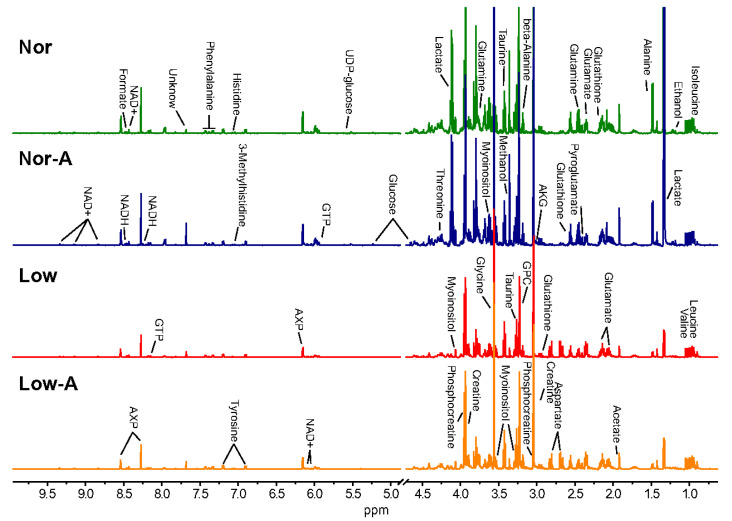
Average 600 MHz 1D ^1^H-NMR spectra recorded on aqueous extracts from the Nor, Nor-A, Low, and Low-A groups of myotubes. The vertical scales were kept constant in all the NMR spectra. The water region (4.7–4.9 ppm) was removed. Green/blue/red/orange lines: the spectral region from the Nor/Nor-A/Low/Low-A groups. Abbreviations: AKG, α-ketoglutarate; GPC, sn-glycerol-3-phosphocholine; UDP-glucose, uridine diphosphate glucose; GTP, guanosine triphosphate; NAD+, nicotinamide adenine dinucleotide; AXP, adenine mono/di/tri phosphate.

**Figure 3 molecules-28-03840-f003:**
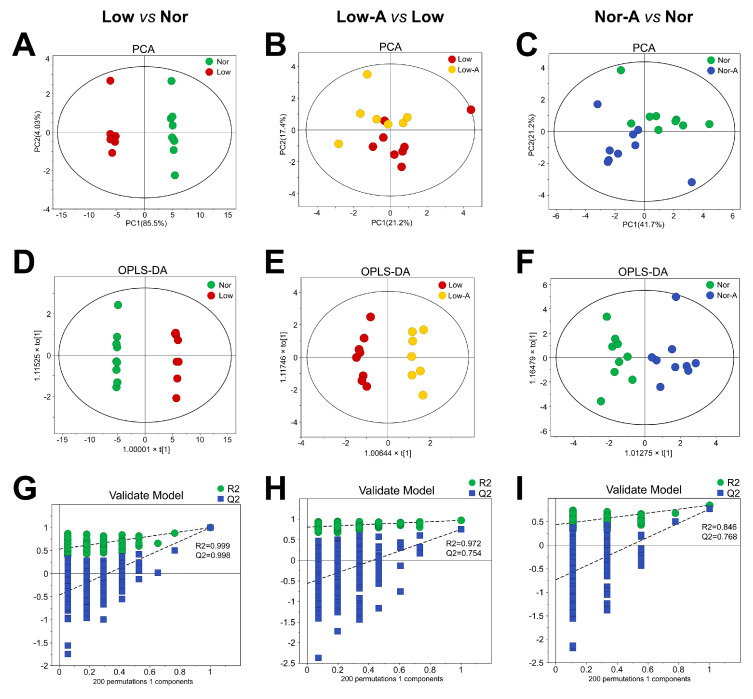
Multivariate statistical analysis for 1D ^1^H-NMR spectra recorded on aqueous extracts from the Nor, Nor-A, Low, and Low-A groups of myotubes. (**A**–**C**) PCA scores plots of Low and Nor, Low-A and Low, and Nor-A and Nor; (**D**–**F**) OPLS-DA scores plots; and (**G**–**I**) OPLS-DA cross-validation plots of Low vs. Nor, Low-A vs. Low, and Nor-A vs. Nor. The ellipses indicate the 95% confidence limits. The cross-validation plots were generated from permutation tests (*n* = 200).

**Figure 4 molecules-28-03840-f004:**
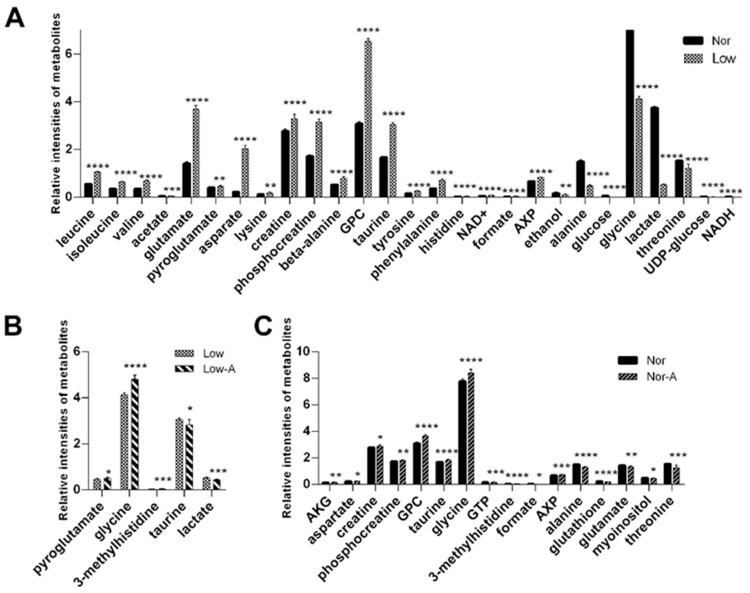
Relative levels of differential metabolites identified from pairwise comparisons between the four groups of myotubes. (**A**) Low vs. Nor; (**B**) Low-A vs. Low; and (**C**) Nor-A vs. Nor. Differential metabolites were identified from Student’s *t*-test analysis with a criterion of *p* < 0.05. Statistical significances: *, *p* < 0.05, **, *p* < 0.01, ***, *p* < 0.001, ****, *p* < 0.0001.

**Figure 5 molecules-28-03840-f005:**
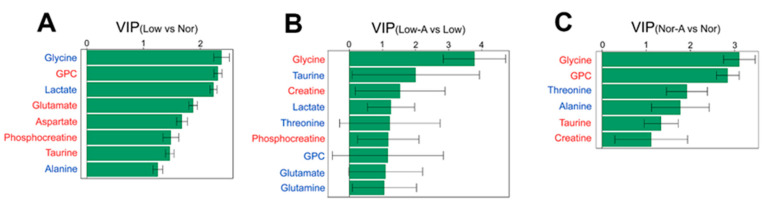
VIP scores of significant metabolites identified from pairwise comparisons between the four groups of myotubes. (**A**) Low vs. Nor; (**B**) Low-A vs. Low; and (**C**) Nor-A vs. Nor. Significant metabolites were identified from the OPLS-DA models with a criterion of VIP > 1. Red/blue font indicates an increased/decreased level of the metabolite.

**Figure 6 molecules-28-03840-f006:**
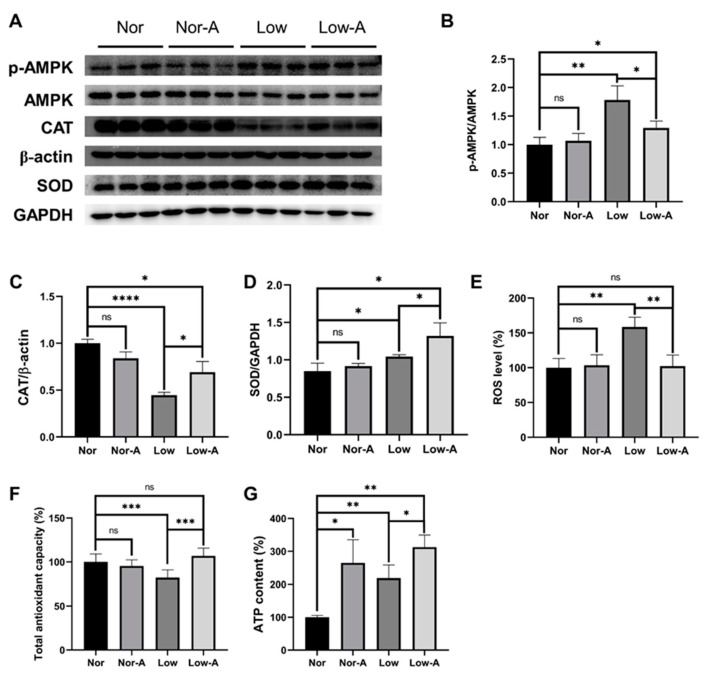
Expressions of proteins corrected with energy metabolism and antioxidant capacities of the four groups of myotubes. (**A**) Western blot analysis of energy metabolism-corrected and antioxidant-corrected proteins. The anti-β-actin antibody and anti-GAPDH proteins were used to standardize amounts of the SOD and CAT proteins, respectively. (**B**) ratio of p-AMPK (T172)/AMPK; (**C**) expressions of the catalase (CAT) protein; (**D**) expressions of the superoxide dismutase (SOD) protein; (**E**) levels of reactive oxygen species (ROS); (**F**) total antioxidant capacities; and (**G**) ATP contents. Statistical significances: ns, *p* > 0.05; *, *p* < 0.05, **, *p* < 0.01, ***, *p* < 0.001, **** *p* < 0.0001. *n* = 4 for each group.

**Figure 7 molecules-28-03840-f007:**
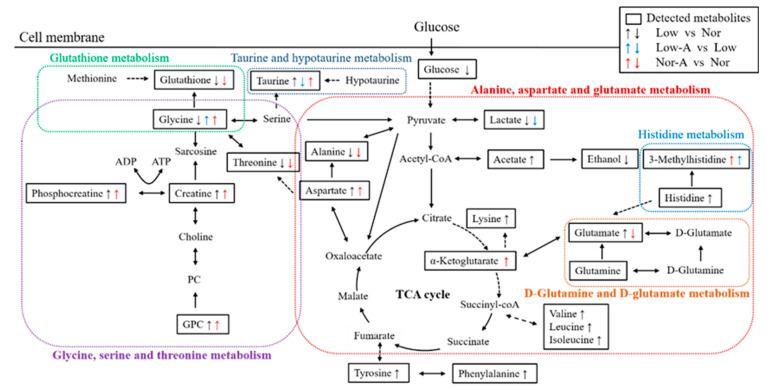
Schematic representation of significantly altered metabolic pathways identified from pairwise comparisons between the four groups of myotubes. The up/down arrow highlights the increased/decreased metabolite; the dotted arrow indicates multiple biochemical reactions; and the solid arrow denotes a single biochemical reaction. Significantly altered metabolic pathways were identified using the KEGG database and the MetaboAnalyst webserver.

**Table 1 molecules-28-03840-t001:** Characteristic metabolites identified from pairwise comparisons between the four groups of myotubes.

Metabolite	Comparison		
	Low vs. Nor	Low-A vs. Low	Nor-A vs. Nor
Alanine	↓	ns	↓
Glutamate	↑	ns	ns
Aspartate	↑	ns	ns
Creatine	ns	ns	↑
Phosphocreatine	↑	ns	ns
GPC	↑	ns	↑
Taurine	↑	↓	↑
Glycine	↓	↑	↑
Lactate	↓	↓	ns
Threonine	ns	ns	↓

Characteristic metabolites were identified with two criteria: *p* < 0.05 obtained from the univariate statistical analysis and VIP > 1 calculated from the OPLS-DA analysis. **↑**/**↓** indicates the increase/decrease of the metabolite; ns: *p* > 0.05.

**Table 2 molecules-28-03840-t002:** Significantly altered metabolic pathways identified from pairwise comparisons between Low and Nor groups, Low-A and Low groups, and Nor-A and Nor groups.

No.	Metabolic Pathway	Low vs. Nor	Low-A vs. Low	Nor-A vs. Nor
P1	Alanine, aspartate, and glutamate metabolism	√		√
P2	Glycine, serine, and threonine metabolism	√	√	√
P3	Glutathione metabolism	√	√	√
P4	D-Glutamine and D-glutamate metabolism	√		√
P5	Beta-Alanine metabolism	√		
P6	Taurine and hypotaurine metabolism	√	√	√
P7	Phenylalanine metabolism	√		
P8	Phenylalanine, tyrosine, and tryptophan biosynthesis	√		
P9	Nicotinate and nicotinamide metabolism	√		
P10	Histidine metabolism	√	√	√

## Data Availability

Data is contained within the article or Appendix A.

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
