# Peer review of "NMR-Based Metabolic Profiling of the Effects of α-Ketoglutarate Supplementation on Energy-Deficient C2C12 Myotubes"

_molecules, 2023, doi:10.3390/molecules28093840_

Round 1
Reviewer 1 Report
In the manuscript entitled "NMR-based metabolic profiling of the effects of α-ketoglutarate supplementation on energy-deficient C2C12 myotubes" Li et al. described the critical role of AKG during the differentiation of myoblasts. They presented some evidence that AKG promotes differentiation and facilitates ROS clearance. In addition, they showed its role in amino acid and antioxidant metabolisms. This is a well-written archive paper that paves the way for further AKG metabolic studies in skeletal muscle.
I have a few concerns:
- In Figure 1, BF pictures are very small, and it is impossible to distinguish myotube formation. The authors should immunostain the cultures with MyHC (MF20), present better representative pictures, and calculate the fusion index and the size of the myotubes.
- How many times the western blots in Figure 6 were repeated? The difference among the groups (westerns) does not appear to be as significant as they are presented in the graphs (calculations). I prefer to see additional western blot analyses that confirm (replicate) the presented results in Figure 6.
Author Response
Q1. In Figure 1, BF pictures are very small, and it is impossible to distinguish myotube formation. The authors should immunostain the cultures with MyHC (MF20), present better representative pictures, and calculate the fusion index and the size of the myotubes.
A1. Thank you for your constructive comment and suggestion. We have regulated the size of panel A in Figure 1 so that it is easy now to observe the morphology of myotubes. Furthermore, Figure S1 provides images at different magnifications to illustrate myotube formation. Your suggestion is very meaningful providing a significant instruction to our experiments, which will be adopted in our future study.
Q2. How many times the western blots in Figure 6 were repeated? The difference among the groups (westerns) does not appear to be as significant as they are presented in the graphs (calculations). I prefer to see additional western blot analyses that confirm (replicate) the presented results in Figure 6.
A2. Thanks for your question and comment. We conducted western blots experiments twice or more times for each experiment. Some of the extra protein bands are showed in the attached PDF file for your reference.

Reviewer 2 Report
The current manuscript from Li et al investigates the metabolic effect of AKG on myotube formation in different energy states by using NMR-based metabolomics. The results show that AKG supplementation promoted myotube formation and induced metabolic remodeling in myotubes, especially, promoting energy metabolism and antioxidant capacity of myotubes. The work explores an interesting topic, and enhances the understanding of the potential molecular mechanisms underlying the influence of AKG on myotube formation. I therefore suggest its publication in the Molecules. But, also please respond to a few questions or comments below.
In Figure2, please replace GDP-lucose with GDP-glucose.
In Table S1, please double check the assignments of threonine, CH3 (1.33 ppm, d); Phosphocreatine, CH3 (3.94 ppm, s).
What difference and connection between Figure 4 and Table 1?
Author Response
Q1. In Figure2, please replace UDP-lucose with UDP-glucose.
A1. Thank you for your kind suggestion. We have fixed this typo in the revised Figure 2.
Q2. In Table S1, please double check the assignments of threonine, CH3 (1.33 ppm, d); Phosphocreatine, CH3 (3.94 ppm, s).
A2. Thanks for your constructive suggestion. We have confirmed the assignments of threonine and phosphocreatine using a combination of 1D 1H-NMR spectrum with 2D NMR spectra, and changed the chemical shift of CH3 in threonine from 1.30 (d) to 1.33 (d), and that of CH3 in phosphocreatine from 4.05 (s) to 3.94 (s), as shown in the revised Table S1 with highlighted revisions.
Q3. What difference and connection between Figure 4 and Table 1?
A3. Thank you for your meaningful question. Figure 4 shows the relative levels of differential metabolites, which were identified with the criterion of p < 0.05 obtained from the univariate statistical analysis. Table 1 shows the characteristic metabolites, which were identified with two criteria of p < 0.05 obtained from the univariate statistical analysis and VIP > 1 calculated from the OPLS-DA analysis.